# Prevalence and Associated Factors of Frailty in Community-Dwelling Older Adults in Indonesia, 2014–2015

**DOI:** 10.3390/ijerph17010010

**Published:** 2019-12-18

**Authors:** Supa Pengpid, Karl Peltzer

**Affiliations:** 1ASEAN Institute for Health Development, Mahidol University, Salaya, Nakhon Pathom 73170, Thailand; supaprom@yahoo.com; 2Lifestyle Diseases Research Entity, Faculty of Health Sciences, North-West University, Mmabatho 2745, South Africa

**Keywords:** frailty, prevalence, correlates, older adults, Indonesia

## Abstract

*Objective*: The investigation aims to study the prevalence and correlates of frailty in a national community-dwelling sample of older Indonesians. *Methods*: Participants were 2630 older adults, 60 years and older (median age 66.0 years, interquartile range = 9.0) who took part in the cross-sectional Indonesia Family Life Survey (IFLS-5) in 2014–2015. They were requested to provide information about sociodemographic and various health variables, including frailty. Multivariable Poisson regression analysis was utilized to estimate the correlates of socio-demographic factors, health variables, and frailty. *Results*: The overall prevalence of frailty was 8.1%; 61.6% were prefrail. In adjusted Poisson regression analysis, older age, being unmarried, separated, divorced or widowed, residing in Java and major island groups, poor cognitive functioning, loneliness, and functional disability were associated with frailty. *Conclusion*: Several sociodemographic and health risk factors for frailty were identified that can help in guiding intervention strategies in Indonesia.

## 1. Introduction

Frailty syndrome can be conceptualized as “a clinical syndrome (phenotype) or as deficits/co-morbidities/disabilities accumulation.” [1]. Using these two concepts, an individual can be classified as “normal (fit/robust), pre-frail, and frail” [2,3]. “Frailty is responsible for enhanced vulnerability to endogenous and/or exogenous stressors, exposing individuals to an increased risk of negative health-related outcomes.” [4]. One of the most common methods to understand frailty as a syndrome in population-based studies is using operational criteria for the frailty phenotype [5,6]. “As a phenotype, frailty syndrome is characterized by exhaustion due to poor endurance and lack of energy, as well as a decrease in body weight (shrinking), muscle strength (weakness), gait speed (slowness), and physical activity” [1,6].

The prevalence of frailty in individuals 60 years and older in low- and middle-income countries was 17.4% and pre-frailty 49.3% [7]. There is scant information on frailty prevalence and correlates in lower-middle-income countries, such as Indonesia [8]. In a study of 448 geriatric clinic patients (60 years and older) in Indonesia, 25.2% were frail [8]. In population-based studies in countries in the East and Southeast Asian region, the prevalence of frailty was 7.0% in China (60 years or older) [9], 20% in India (50 years and older) [10], 9.4% in Malaysia (60 years and older) [11], and 5.7% in Singapore (60 years and older) [12]. There is a need to estimate the national population-based prevalence of frailty and its associated factors in Indonesia. Understanding the profile of frailty will help in assessing the impact of services and policies for the prevention and control of frailty in the future.

Risk factors for frailty include sociodemographic and health related variables. Sociodemographic risk factors for frailty include, older age [9,12,13,14], female sex [9], lower income [14,15], lower education [9,15], and low social support or poor social networks [12,14]. Health variable risk factors for frailty, may include, poor health status [14,16], low quality of life or low life satisfaction [17,18], poor mental health (depression, insomnia) [19,20], lower cognitive functioning [11,16], higher comorbidity or number of chronic conditions [9,13,14,16,21], and functional disability [9,13,16,17,21]. In addition, behavioral risk factors include, lower fruit and vegetable consumption [22], tobacco use [23], polypharmacy [14,23], and falls [9,14,17].

The study aims to investigate the prevalence and correlates of frailty in a national community-dwelling sample of older persons (60 years and above) who took part in the Indonesian Family Life Survey (IFLS)-5 in 2014–2015.

## 2. Method

### 2.1. Study Design and Participants

Data were analyzed from the cross-sectional Indonesia Family Life Survey (IFLS-5) [24]. Data from the IFLS-5 is available from RAND at http://www.rand.org/labor/FLS/IFLS.html. The IFLS-5 was a population-based household survey conducted in 2014–2015, using a multi-stage stratified sampling design (321 enumeration areas-EAs, 20 and 30 randomly selected households from each urban and rural EA, respectively, in 13 of 27 Indonesian provinces), representing 83% of the Indonesian population [24,25,26]. The computer-assisted personal interview system (CAPI) was utilized for conducting interviews, and the data were entered using CSPro [14,15,16,17,18,19,20,21,22,23,24,25,26,27]. The questionnaire was developed in English and initially translated into Bahasa Indonesia by survey staff and then retranslated into English by two independent, outside translators, and pre-tested on 393 household members [24,25,26]. Ethics review boards of RAND and the University of Gadjah Mada in Indonesia approved the IFLS [24]. In all, after written, informed consent was obtained, 2630 individuals 60 years and older were included with complete frailty measurements. The study response rate was above 90% [24,25,26].

### 2.2. Measures

Measures of frailty in population-based studies may be by self-report, objective, and mixed frailty measures (instruments including both subjective and objective components) [5]. The Fried’s scale [6], a mixed frailty measure, has been the most extensively tested for its validity and is the most widely used instrument in frailty research, allowing comparisons to be made between studies [5]. Predictive validity for the Fried’s scale has been established for mortality, fractures, falls, ADL and IADL, etc. [5].

*Frailty* was assessed in this study using Fried phenotype of frailty: “(1) shrinking or unintentional weight loss, (2) self-reported exhaustion, (3) weakness, (4) slowness, and (5) low physical activity”; weakness and slowness were assessed using objective tests [6]. Each component was scored with zero or one, with 0 scores indication robustness, 1–2 scores prefail, and 3–5 as frail [6]. For the assessment of *shrinking or unintentional weight loss*, heights and weights were taken using standard procedures [24], and body mass index (BMI) was calculated for both sexes in the lowest quintile or a BMI of <18.5 kg/m^2^ as an indication of shrinking [10]. For the assessment of *exhaustion*, 2 items (“I felt that everything I did was an effort”, and “I could not get going.”) from the Centres for Epidemiologic Studies Depression Scale (CES-D: 10 items) were used [28]. Respondents who answered occasionally or most of the time on either of the two questions were categorized as frail by the exhaustion criterion [6]. *Weakness* was assessed with hand grip strength (HGS), using a Baseline Smedley Spring type dynamometer (calibrated daily), on each hand twice, using an HGS (kg) from all four measurements [24,29]. The Smedley dynamometer records measurements to the nearest 0.5 kg of force [30]. The Pearson correlation between forces recorded with the Smedley dynamometer and known forces was 98 [31]. Low HGS was classified as the gender-specific and BMI adjusted the lowest quintile [6,10]. *Slowness* was classified as the lowest height, and sex-adjusted quintile for a 4-m timed walk (average of two walks) [24]. *Low physical activity* was classified according to a brief version of the International Physical Activity Questionnaire (IPAQ) (IPAQ-S7S). [32,33].

Measures for independent variables are summarized in Table 1.

### 2.3. Data Analysis

Bivariate correlations between independent variables and the dependent variable (frailty) were evaluated with Poisson regression calculating prevalence ratios (PR). The dependent variable was dichotomized into 0–2 = 0 no frailty and 3–5 = 1 frailty. In analyses of data from cross-sectional studies, “Poisson models with robust variance are better alternatives than logistic regression is.” [40]. Variables associated with frailty (age, marital status, formal education, economic status, residence status, region, life satisfaction, subjective health status, cognition, insomnia, loneliness, and functional disability) at *p* < 0.05 were included in a multivariable Poisson regression model. *p* < 0.05 was considered significant. Cross-section analysis weights were applied “to make the IFLS-5 sample representative of the 2014 Indonesian population in the study provinces.” [24,25,26]. Both the 95% confidence intervals and *p*-values were adjusted, taking the complex survey design of the study into account. All analyses were performed using STATA software version 14.0 (Stata Corporation, College Station, TX, USA).

## 3. Results

### 3.1. Sample Characteristics and Frailty

The total sample included 2630 older adults, 60 years and older (median age 66.0 years, interquartile range = 9.0, age range of 60 to 101 years) in Indonesia. The proportion of women was 49.7%, 75.6% had no or primary education, 40.7% as having medium economic status, 53.1% lived in urban areas, and 59.2% in Java. Almost one in three of the older adults (32.8%) rated their health status as unhealthy, 20.9% had low social capital, and 17.8% had low life satisfaction. The overall prevalence of frailty was 8.1%. In bivariate analysis, older age, being unmarried/separated/divorced/widowed, lower education, poorer economic background, rural residence, residing in Java and major island groups, low life satisfaction, unhealthy subjective health status, poor cognitive functioning, insomnia symptoms, loneliness, and functional disability were positively associated with frailty (see Table 2).

Table 2 shows the prevalence of frailty components and types. Physical inactivity (47.9%) was the most common and exhaustion (12.6%), the least common frailty component. Women were more physically inactive than men, while there were no sex differences for the other four frailty components. Regarding the frailty type, 7.7% of men and 8.5% of women were frail, and 59.5% of men and 63.7% of women were prefrail. Frailty status did not significantly differ between the sexes (See Table 3).

### 3.2. Associations with Frailty

In adjusted Poisson regression analysis, older age, being unmarried, separated, divorced or widowed, residing in Java and major island groups, poor cognitive functioning, loneliness, and functional disability were positively associated with frailty (see Table 4).

## 4. Discussion

The study aimed to assess frailty and its correlates in community-dwelling older adults in Indonesia. The prevalence of frailty was 8.1%, which is similar to population-based studies in the region, e.g., China (7.0%) [9], Malaysia (9.4%) [11], and Singapore (5.7%) [12], but lower than the global prevalence in low- and middle-income countries (17.4%) [7]. A higher prevalence of frailty was found among geriatric clinic patients in Indonesia (25.2%) [8], which may be explained by the differences in the recruitment setting; a higher prevalence of frailty is expected in geriatric clinic patients at referral hospitals than in a community setting. Physical inactivity was the most common frailty component in this study, which compares with a study in Germany [14] and demonstrates the importance of the promotion of physical activity in this population.

In consistence with previous studies [9,12,13,14,15], this study found that older age, being unmarried, separated, divorced or widowed, and in the bivariate analysis that poorer economic status and lower education were associated with frailty. Frailty is associated with aging by accumulating deficiencies in several physiological systems [21]. The importance of partner support in relation to frailty, emphasizes the need for social support [21]. Persons with lower education and lower income may access health services and practice health behaviors less often, which may contribute to the development of frailty [10,21]. Compared to Sumatra, the prevalence of frailty was higher in the other study regions (Java, Major island groups). This finding will need further research. While some previous studies [12,14] found an association between low social support or poor social networks and frailty, this study did not find such an association. However, it is possible that multiple social factors from a range of spheres of influence (e.g., individual, family, peer group, neighborhood, and society) interact with one another to impact health issues, such as frailty [41].

This study found in a bivariate analysis that poor health status and low life satisfaction were associated with frailty. This result is consistent with previous studies [14,16,17,18]. Older adults with poor self-rated health and/or low life satisfaction may limit investment in self-care and health behaviors, such as physical activity, dietary, and health-seeking behavior, and may thus, develop more likely frailty [21].

In agreement with other studies [19,20], this study also found that poor mental health (loneliness and in the bivariate analysis, insomnia) was associated with frailty. Poor cognitive functioning was in agreement with previous studies [11,16] associated with frailty. Some research indicates poor cognition and frailty share “underlying biological causal explanatory factors” [16]. A number of previous studies [9,13,14,16,21] found an association between higher comorbidity or number of chronic conditions and frailty, while this study did not find any significant association. The finding that frailty “can occur in the absence of multimorbidity” has also been found in a study in Brazil [21]. In line with a number of studies [9,13,16,17,21], this study found an association between functional disability and frailty. This shows that our study frailty has overlap with function disability but not comorbidity. The link between frailty, physical activity level and functional disability assessed by means of the ADL and IADL questionnaire may be explained by, for example, that frail individuals reported different levels of mobility function compared to no frail older adults [42] and that “frail elders, men, those who are older, overweight or have multiple comorbidities are most likely to have low activity” [43]. Several behavioral risk factors (lower fruit and vegetable consumption, tobacco use, and falls) [9,14,17,22,23]. have been found correlated with frailty, while this study did not find any correlation.

### Limitations of the Study

The study was limited by the cross-sectional nature of the study. Further, the survey methodology focused on community-dwelling older adults, and by doing so, excluded institutionalized older adults that could have had a higher frailty prevalence. A further limitation was that the IPAQ [32,33] is used in populations 15–69 years, and in this study, persons 70 years and older were included. In a validation study in Japan, the IPAQ was found a useful tool (adequate validity) for assessing physical activity among older adults [44]. However, some studies (e.g., [45]) suggest using a modified IPAQ for the elderly, which should be considered in future studies. Further limitations include the measurement of certain concepts with single items, such as some aspects of frailty, life satisfaction, and self-reported health status.

## 5. Conclusions

The study found that the prevalence of frailty in individuals 60 years and older in Indonesia is similar to that found in corresponding age groups in several East and Southeast Asian countries.

Several sociodemographic and health risk factors, such as older age, living without a partner, poor cognitive functioning, loneliness, and functional disability were identified for frailty that can guide intervention strategies and the implementation of health care provision that can, in turn, promote active aging in Indonesia.

## Figures and Tables

**Table 1 ijerph-17-00010-t001:** Independent study variables included in this investigation.

Measure	Questions	Response Options	Classification
Socio-demographic variables [24,25,26]	How old are you?	Age in years	60–69 70–79 80 or more
	Sex	Male, female	Male, female
	Residential status	Urban, rural	
	Country region	Sumatra, Java, and Major island groups (Bali, West Nusa Tenggara, South Kalimantan, and South Sulawesi)	Sumatra, Java, and Major island groups
	Education	None, elementary, high school, higher education	Low = None to high school and high = higher education
	Marital status	Never married, married, separated, divorced, widow/er, cohabitate	Married/cohabiting = 0 and Never married, separated, divorced, widow/er = 1
Subjective socioeconomic status [24]	“Please, imagine a six-step ladder where on the bottom (the first step) stand the poorest people, and on the highest step (the sixth step) stand the richest people. On which [economic] step are you today?”	The answers ranged from (1) poorest to (6) richest	Economic step 1 to 2 was classified as poor, 3 as medium and 4 to 6 as rich economic status
Social capital [24,25,26]	Four questions on past 12-month participation in four different community activities (Cronbach’s alpha 0.69)	Yes/No	Low social capital was defined as having not participated in any community activities
Life satisfaction [24]	“Please, think about your life as a whole. How satisfied are you with it?”	1 = completely satisfied to 5 = not all satisfied	Low life satisfaction was defined as not very or not at all satisfied
Self-reported health status [24,25,26]	“In general, how is your health?”	Response options were ranged from 1 = Very healthy to 4 = Unhealthy	Very healthy/Somewhat healthy = 0 and Somewhat unhealthy/Unhealthy = 1
Cognitive functioning [24,34]	Questions from the telephone survey of cognitive status (TICS)	Total scores of the TICS ranged from 0–34	A score of 13 or less was defined as low
Insomnia symptoms	Five items from the Patient-Reported Outcomes Measurement Information System (PROMIS) sleep disturbance measure [35] and with five items from the PROMIS sleep impairment measure [36] (Cronbach’s alpha was 0.82)	1 = Never/Not at all to 5 = Very much/Always	Insomnia was defined as having total scores of ≥21–40 [37]
Loneliness	One item from the “Center for Epidemiologic Studies Depression Scale” (CES-D-10): “How often did you feel lonely in the past week?” [28].	1 = Rarely or none (≤1 day) to 4 = Most of the time (5–7 days)	Loneliness was defined as occasionally or all of the time or 3–7 days in the past week lonely
Infrequent fruit and vegetable consumption [24,26]	Questions on the number of days in the past week vegetables (green leafy vegetables and carrots) and fruits (banana, papaya, and mango) had been consumed.	1–7 days	Eating less than 3 days a week fruits and less than daily vegetables
Current tobacco use [24]	“Have you ever chewed tobacco, smoked a pipe, smoked self-enrolled cigarettes, or smoked cigarettes/cigars?” “Do you still have the habit, or have you totally quit?”	Yes, No Still have, Quit	Never, former Current
Chronic condition [24,27]	Health care provider diagnosed 15 different types of illnesses, e.g., diabetes or high blood sugar and arthritis/rheumatism	Yes/No	None One Two or more
Functional disability [25,38,39]	Five items of Activity of Daily Living (ADL) (Cronbach alpha 0.84) and six items of Instrumental Activity of Daily Living (IADL) (Cronbach alpha 0.91)	1 = Easily to 4 = Unable to do it	The total functional disability score was classified into 0 = having no difficulty, 1 = one, and 2 = two or more ADL/IADL items.
Falls [24]	“Have you fallen down in the last two years and received treatment?”	Yes/No	No = 0 and Yes = 1

**Table 2 ijerph-17-00010-t002:** Sample characteristics and prevalence of frailty among older adults in Indonesia.

Variables		Total Sample	Frailty	Bivariate Analysis
		*N* (%)		PR (95% CI)
All Age in years	60–69 70–79 80 and over	2630 1784 (70.3) 725 (25.3) 121 (4.4)	214 (8.1) 83 (4.7) 99 (14.2) 32 (27.4)	1 (Reference) 3.02 (2.22, 4.11) *** 5.88 (3.96, 8.73) ***
Gender	Female Male	1330 (49.7) 1300 (50.3)	105 (8.5) 109 (7.7)	1 (Reference) 0.91 (0.68, 1.21)
Marital status	Married/cohabiting Unmarried/separated/divorced/widowed	1733 (67.8) 897 (32.2)	117 (6.3) 97 (11.9)	1 (Reference) 1.89 (1.42, 2.51) ***
Formal education	Low High	1947 (75.6) 675 (24.4)	185 (9.4) 29 (3.9)	1 (Reference) 0.42 (0.28, 0.64) ***
Economic background	Poor Medium Rich	863 (32.2) 1080 (40.7) 687 (26.0)	83 (10.2) 82 (7.6) 49 (6.2)	1 (Reference) 0.74 (0.54, 1.02) 0.61 (0.42, 0.89) *
Residence	Rural Urban	1233 (46.9) 1397 (53.1)	118 (9.5) 96 (6.5)	1 (Reference) 0.69 (0.52, 0.91) **
Region	Sumatra Java Major island groups	527 (20.2) 1557 (59.2) 546 (20.8)	31 (5.1) 132 (8.4) 51 (9.3)	1 (Reference) 1.67 (1.13, 2.47) ** 1.83 (1.18, 2.86) **
Social capital	High Low	2058 (79.1) 572 (20.9)	157 (7.7) 57 (9.4)	1 (Reference) 1.22 (0.88, 1.69)
Life satisfaction	Moderate/High Low	2153 (82.2) 417 (17.8)	162 (7.4) 52 (11.1)	1 (Reference) 1.49 (1.07, 2.06) *
Subjective health status	Healthy Unhealthy	1703 (67.2) 927 (32.8)	118 (7.0) 96 (10.3)	1 (Reference) 1.47 (1.10, 1.96) **
Cognition	High Low	1555 (64.1) 875 (35.9)	59 (4.0) 81 (11.7)	1 (Reference) 2.92 (2.05, 4.18) ***
Insomnia	No Yes	2361 (90.0) 268 (10.0)	179 (7.5) 35 (12.8)	1 (Reference) 1.68 (1.15, 2.45) **
Lonely	No Yes	2481 (94.2) 149 (5.8)	190 (7.5) 24 (16.9)	1 (Reference) 2.26 (1.49, 3.43) ***
Fruit and vegetable consumption	Frequent Infrequent	1813 (67.1) 816 (32.9)	133 (7.4) 81 (9.5)	1 (Reference) 1.29 (0.96, 1.73)
Tobacco use status	Never, former Current	1764 (66.0) 866 (34.0)	146 (8.3) 68 (7.7)	1 (Reference) 0.92 (0.68, 1.25)
Chronic conditions	None One Two or more	1336 (52.7) 718 (26.9) 576 (20.4)	103 (7.6) 58 (8.4) 53 (9.1)	1 (Reference) 1.10 (0.79, 1.55) 1.21 (0.84, 1.72)
ADL/IADL	None One Two or more	1827 (70.5) 605 (22.9) 198 (6.7)	107 (5.8) 69 (10.9) 38 (22.0)	1 (Reference) 1.87 (1.35, 2.57) *** 3.79 (2.63, 5.46) ***
Fall past 2 years	No Yes	2330 (88.6) 299 (11.4)	184 (8.0) 30 (8.5)	1 (Reference) 1.06 (0.70, 1.61)

PR = Prevalence Ratio; *** *p* < 0.001; ** *p* < 0.01; * *p* < 0.05; (I) ADL = (Instrumental) Activities of Daily Living.

**Table 3 ijerph-17-00010-t003:** Prevalence of frailty components and types by gender.

	Total (*N* = 2630) % (CI)	Men (*N* = 1300) % (CI)	Women (*N* = 1330) % (CI)
*Frailty components*			
Exhaustion	12.6 (11.4, 14.0)	12.6 (10.8, 14.7)	12.7 (10.9, 14.6)
Low body weight	19.9 (18.5, 21.4)	20.1 (18.8, 23.0)	19.2 (17.3, 21.2)
Low physical activity	47.9 (45.9, 49.9)	42.7 (39.9, 45.6)	52.3 (50.0, 55.6)
Slowness	20.0 (18.6, 21.6)	20.0 (17.9, 22.3)	20.0 (18.0, 22.2)
Low grip strength	19.3 (17.9, 20.8)	18.1 (16.1, 20.2)	20.5 (18.4, 22.7)
*Frailty type*			
Robust	30.3 (28.4, 32.3)	32.8 (30.0, 35.7)	27.8 (25.2, 30.6)
Prefrail	61.6 (60.0, 63.7)	59.5 (56,5, 62.4)	63.7 (60.8, 66.5)
Frail	8.1 (7.0, 9.3)	7.7 (6.3, 9.4)	8.5 (6.9, 10.3)

**Table 4 ijerph-17-00010-t004:** Multivariable Poisson regression analysis of factors associated with frailty among older adults in Indonesia.

Variables	PR (95% CI)	*p*-Value
Age in years 60–69 70–79 80 and over	1 (Reference) 2.15 (1.47, 3.13) 3.95 (2.60, 5.98)	<0.001 <0.001
Marital status Married/cohabiting Unmarried/separated/divorced/widowed	1 (Reference) 1.43 (1.07, 1.91)	0.017
Formal education Low High	1 (Reference) 0.68 (0.43, 1.05)	0.083
Economic background Poor Medium Rich	1 (Reference) 0.96 (0.69, 1.32) 0.88 (0.60, 1.29)	0.809 0.515
Residence Rural Urban	1 (Reference) 0.90 (0.67, 1.20)	0.167
Region Sumatra Java Major island groups	1 (Reference) 1.87 (1.14, 3.08) 2.04 (1.16, 3.58)	0.014 0.013
Life satisfaction Moderate, high Low	1 (Reference) 1.12 (0.80, 1.56)	0.522
Subjective health status Healthy Unhealthy	1 (Reference) 1.27 (0.95, 1.68)	0.105
Cognition High Low	1 (Reference) 2.12 (1.46, 3.07)	<0.001
Insomnia No Yes	1 (Reference) 1.30 (0.88, 1.92)	0.192
Lonely No Yes	1 (Reference) 1.74 (1.16, 2.60)	0.008
Functional disability ADL/IADL = 0 ADL/IADL = 1 ADL/IADL = 2 or more	1 (Reference) 1.57 (1.15, 2.14) 2.41 (1.65, 3.52)	0.005 <0.001

PR = Prevalence Ratio; (I) ADL = (Instrumental) Activities of Daily Living.

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
