# Peer review of "Prevalence and Associated Factors of Frailty in Community-Dwelling Older Adults in Indonesia, 2014–2015"

_ijerph, 2019, doi:10.3390/ijerph17010010_

Round 1
Reviewer 1 Report
This study aims to study the prevalence and impact factors of frailty of community-dwelling older in Indonesians with a nationally representative sample. However, some key problems must be resolved before its publication, especially for the method section.
Introduction
1.Literatures review must be strengthened, especially previous studies related frailty of in Indonesia elderly. (such as Setiati S, Laksmi PW, Aryana IS, Sunarti S, Widajanti N, Dwipa L, Seto E, Istanti R, Ardian LJ, Chotimah SC. Frailty state among Indonesian elderly: prevalence, associated factors, and frailty state transition. BMC geriatrics. 2019 Dec;19(1):182.)
2. The the current situation and trends of the measurement instruments for frailty should be introduced and compared briefly.
3. The difference between this study to previous studies, and the significance of this study should be clarified in this section.
Methodology
4. The survey “Indonesia Family Life Survey (IFLS-5)” should be introduced, especially the setting, sampling, survey process and quality control of the survey.
5. The frailty measurement instruments used in this study should be introduced in details and its characteristics compared to previous studies. The validity of the instruments should be described.
6. As the authors described, the frailty are measured by five binary variables at individual levels. How the five variables were combined to measure the frailty should be described clearly. In addition, may be the Poisson regression method are not suitable for this study if the dependent variable represents the times of the five questions values are one, the dependent variable- times of value – not follow the Poisson distribution for the five questions are correlated at individual level, the incidence rate is not vary low as showed in the results, and the occurred time or space are not specified. Overall, the authors must demonstrate the dependent variable follow the Poisson distribution. Otherwise, the model must be replaced by other model.
7. The variables values should be introduced clearly, the authors could use a table to show the definitions of questions and variables.
8. The model formula for estimating the relationship between the dependent and independent variables should be presented.
Results and discussion
9. Sample characteristics are not the emphasis of this study, therefore, the description about the sample characteristics should be simplified. Only the important characteristics related to frailty should be described.
10. line 120-123, the relationship direction should be proposed. “The overall prevalence of frailty was 8.1%. In bivariate analysis, older age, being unmarried/separated/divorved/widowed, lower education, lower economic background, rural residence, low life satisfaction, unhealthy subjective health status, poor cognitive functioning, insomnia symptoms, loneliness, and functional disability were associated with frailty (see Table 1).” Similar problems for line 135-137.
11. The discussion should focus on community-dwelling older adults, and compared with others population, for example in institutions. Highlight the unique factors related to frailty in community-dwelling older adults.
Minor suggestions:
Line 26 “[1]_The global” the bottom line after [1] should be deleted.
Line 172-173, the fonts size are different.
Title: the “correlates” may be replaced by much more proper words, which indicates the related factors of frailty.
Author Response
Reviewer 1
Comments and Suggestions for Authors
This study aims to study the prevalence and impact factors of frailty of community-dwelling older in Indonesians with a nationally representative sample. However, some key problems must be resolved before its publication, especially for the method section.
Introduction
1.Literatures review must be strengthened, especially previous studies related frailty of in Indonesia elderly. (such as Setiati S, Laksmi PW, Aryana IS, Sunarti S, Widajanti N, Dwipa L, Seto E, Istanti R, Ardian LJ, Chotimah SC. Frailty state among Indonesian elderly: prevalence, associated factors, and frailty state transition. BMC geriatrics. 2019 Dec;19(1):182.)
Response: this study was already included, as in below, but now the updated reference is included
In a study of 448 geriatric clinic patients (60 years and older) in Indonesia 25.2% were frail [3].
Further, below is now included
Laksmi, P.W. Frailty syndrome: an emerging geriatric syndrome calling for its potential intervention. Acta Med Indones. 2014;46(3):173-4.
2. The the current situation and trends of the measurement instruments for frailty should be introduced and compared briefly.
Response: below is added
Frailty syndrome, can be conceptualized as “a clinical syndrome (phenotype) or as deficits/co-morbidities/disabilities accumulation.”[1] Using these two concepts, an individual can be classified as “normal (fit/robust), pre-frail, and frail”[2,3]. “Frailty is responsible for enhanced vulnerability to endogenous and/or exogenous stressors, exposing individuals to an increased risk of negative health-related outcomes.”[4] One of the most common methods to understand frailty as a syndrome in population-based studies is using operational criteria for the frailty phenotype [5,6]. “As a phenotype, frailty syndrome is characterized by exhaustion due to poor endurance and lack of energy, as well as decrease in body weight (shrinking), muscle strength (weakness), gait speed (slowness), and physical activity”[1,6].
3. The difference between this study to previous studies, and the significance of this study should be clarified in this section.
Response: below is added
There is a need to estimate the national population-based prevalence of frailty and its associated factors in Indonesia. Understanding the profile of frailty will help in assessing the impact of services and policies for the prevention and control of frailty in the future.
Methodology
4. The survey “Indonesia Family Life Survey (IFLS-5)” should be introduced, especially the setting, sampling, survey process and quality control of the survey.
Response: below is added
Data were analysed from the cross-sectional “Indonesia Family Life Survey (IFLS-5)” [24]. The IFLS-5 was a population-based household survey conducted in 2014-2015, using a multi-stage stratified sampling design (321 enumeration areas-EAs, 20 and 30 randomly selected households from each urban and rural EA, respectively, in 13 of 27 Indonesian provinces), representing 83% of the Indonesian population [24-26]. “Computer-assisted personal interview system (CAPI)” was utilized for conducting interviews, and the data were entered using CSPro [14-27]. The “questionnaire was developed in English and initially translated into Bahasa Indonesia by survey staff and then retranslated into English by two independent, outside translators,” and pre-tested on 393 household members [24-26]. Ethics review boards of RAND and the University of Gadjah Mada in Indonesia approved the IFLS [24]. In all, after written informed consent was obtained, 2630 individuals 60 years and older were included with complete frailty measurements. The study response rate was above 90% [24-26].
5. The frailty measurement instruments used in this study should be introduced in details and its characteristics compared to previous studies. The validity of the instruments should be described.
Response: below is added
Measures of frailty in population-based studies may be by self-report, objective, and mixed frailty measures (instruments including both subjective and objective components) [5]. The Fried’s scale [6], a mixed frailty measure, has been the most extensively tested for its validity and is the most widely used instrument in frailty research, allowing comparisons to be made between studies [5]. Predictive validity for the Fried’s scale has been established for mortality, fractures, falls, ADL & IADL, etc. [5].
6. As the authors described, the frailty are measured by five binary variables at individual levels. How the five variables were combined to measure the frailty should be described clearly. In addition, may be the Poisson regression method are not suitable for this study if the dependent variable represents the times of the five questions values are one, the dependent variable- times of value – not follow the Poisson distribution for the five questions are correlated at individual level, the incidence rate is not vary low as showed in the results, and the occurred time or space are not specified. Overall, the authors must demonstrate the dependent variable follow the Poisson distribution. Otherwise, the model must be replaced by other model.
Response: below is added to show the binary dependent variable
The dependent variable was dichotomized into 0-2=0 no frailty and 3-5=1 frailty
7. The variables values should be introduced clearly, the authors could use a table to show the definitions of questions and variables.
Response: This is added, as below
Measures for independent variables are summarized in Table 1.
Table 1. Independent study variables included in this investigation
Measure Questions Response options Classification
Socio-demographic variables [19-21]
How old are you?
Age in years
60-69
70-79
80 or more
Sex Male, female Male, female
Residential status Urban, rural
Country region Sumatra, Java and Major island groups (Bali, West Nusa Tenggara, South Kalimantan, and South Sulawesi) Sumatra, Java and Major island groups
Education None, elementary, high school, higher education Low=None to high school and high=higher education
Marital status Never married, married, separated, divorced, widow/er, cohabitate Married/cohabiting=0 and Never married, separated, divorced, widow/er=1
Subjective socioeconomic status [19] “Please, imagine a six step ladder where on the bottom (the first step) stand the poorest people, and on the highest step (the sixth step) stand the richest people. On which [economic] step are you today?“ The answers ranged from (1) poorest to (6) richest Economic step 1 to 2 was
classified as poor, 3 as medium and 4 to 6 as rich ecomonic status
Social capital [19-21]
Four questions on past 12 month participation in four different community activities (Cronbach’s alpha 0.69), Yes/No Low social capital was defined as having not participated in any community activities
Life satisfaction [19]
“Please, think about your life as a whole. How satisfied are you with it?” 1=completely satisfied to 5=not all satisfied Low life satisfaction was defined as not very or not at all satisfied.
Self-reported health status [19-21]
“In general, how is your health?” Response options were ranged from 1=Very healthy to 4=Unhealthy Very healthy/Somewhat healthy=0 and Somewhat unhealthy/Unhealthy=1
Cognitive functioning [19,28]
Questions from the “Telephone Survey of Cognitive Status (TICS)” Total scores of the TICS ranged from 0-34 13 or less scores was defined as low
Insomnia symptoms Five items from the “Patient-Reported Outcomes Measurement Information System (PROMIS)” sleep disturbance measure [29] and with five items from the PROMIS sleep impairment measure [30] (Cronbach’s alpha was 0.82). 1=Never/Not at all to 5=Very much/Always Insomnia was defined as having total scores of ≥21-40 [31]
Loneliness
One item from the “Center for Epidemiologic Studies Depression Scale (CES-D-10): “How often did you feel lonely in the past week?”[24] 1=Rarely or none (≤ 1 day) to 4=Most of the time (5-7 days) Loneliness was defined as occasionally or all of the time or 3-7 days in the past week lonely.
Infrequent fruit and vegetable consumption [19,21]
Questions on the number of days in the past week vegetables (green leafy vegetables and carrots) and fruits (banana, papaya and mango) had been consumed. 1-7 days Eating less than 3 days a week fruits and less than daily vegetables
Current tobacco use [19]
“Have you ever chewed tobacco, smoked a pipe, smoked self-enrolled cigarettes, or smoked cigarettes/cigars?”
“Do you still have the habit or have you totally quit?” Yes, No
Still have, Quit Never, former
Current
Chronic condition [19,22]
Health care provider diagnosed 15 different types of illnesses, e.g., diabetes or high blood sugar and arthritis/rheumatism Yes/No None
One
Two or more
Functional disability [20,32,33]
Five items of “Activity of Daily Living (ADL)” (Cronbach alpha 0.84) and six items of “Instrumental Activity of Daily Living (IADL)” (Cronbach alpha 0.91) 1=Easily to 4=Unable to do it The total functional disability score was classified into 0=having no difficulty, 1=one, and 2=two or more ADL/IADL items.
Falls [19]
“Have you fallen down in the last two years and received treatment?” Yes/No No=0 and Yes=1
8. The model formula for estimating the relationship between the dependent and independent variables should be presented.
Response: added as below
The dependent variable was dichotomized into 0-2=0 no frailty and 3-5=1 frailty. Variables associated with frailty (age, marital status, formal education, economic status, residence status, region, life satisfaction, subjective health status, cognition, insomnia, loneliness and functional disability) at P<0.05 were included in a multivariable Poisson regression model.
Results and discussion
9. Sample characteristics are not the emphasis of this study, therefore, the description about the sample characteristics should be simplified. Only the important characteristics related to frailty should be described.
Response: changed accordingly
10. line 120-123, the relationship direction should be proposed. “The overall prevalence of frailty was 8.1%. In bivariate analysis, older age, being unmarried/separated/divorved/widowed, lower education, lower economic background, rural residence, low life satisfaction, unhealthy subjective health status, poor cognitive functioning, insomnia symptoms, loneliness, and functional disability were associated with frailty (see Table 1).” Similar problems for line 135-137.
Response: Corrected accordingly
11. The discussion should focus on community-dwelling older adults, and compared with others population, for example in institutions. Highlight the unique factors related to frailty in community-dwelling older adults.
Response: below is added
The study aimed to assess frailty and its correlates in community-dwelling older adults in Indonesia. The prevalence of frailty was 8.1%, which is similar to population-based studies in the region, e.g. China (7.0%) [9], Malaysia (9.4%) [11], and Singapore (5.7%) [12], but lower than the global prevalence in low- and middle-income countries (17.4%) [7].
A higher prevalence of frailty was found among geriatric clinic patients in Indonesia (25.2%) [8], which may be explained by the differences in the recruitment setting; a higher prevalence of frailty is expected in geriatric clinic patients at referral hospitals than in a community setting.
Minor suggestions:
Line 26 “[1]_The global” the bottom line after [1] should be deleted.
Response: deleted
Line 172-173, the fonts size are different.
Response: Corrected
Title: the “correlates” may be replaced by much more proper words, which indicates the related factors of frailty.
Response: Corrected
Reviewer 2 Report
The aim of this brief study was to evaluate the prevalence of frailty in Indonesian older adults as well as to describe the factors associated with the frailty. Here are my comments to the authors.
Abstract
I suggest to include the prevalence also of the data about pre-frailty people
Line 27
Please delete “_”
Lines 45-50
I suggest to include the ethical approvals in the paper rather as citation.
Lines 61-62
What type of Hand grip? Please add this information as well as the accuracy of the instrument
Line 65-66
IPAQ: the authors used the IPAQ to investigate PA levels. However the IPAQ is valid to assess PA in adults and not in older adults (aged over 60 years). Please discuss this aspect as limitation of the study. It is possible that the data may have a bias.
Line 121
The data about region is missing
it is divorced not divorved.
Economic background: based on table 1 the information is different. Please check “lower economic background”. The difference is between Poor and rich category.
Is lower economic background the same of poor in table 1? I suggest to use the same words. Please check all.
Line 142-143
What is the difference with the higher prevalence of the study conducted in Indonesia (reference 4)?
Please discuss and explain this aspect.
Line 171-172
I suggest to better explain the link between frailty, physical activity level and functional disability assessed by means of the ADL and IADL questionnaire. For example, it was demonstrated that frail individuals reported different levels of mobility function compared to no frail older adults (PMID: 30898096). Again a recent study suggested that frail elders, men, those who are older, overweight or have multiple comorbidities are most likely to have low activity (PMID: 29106478). I think that this is an important aspect of frailty and I suggested to the author to deepen this topics.
Lines 175-178
Please explicit in the limitation of the study the use of single item to investigate some aspects about frailty (e.g., life satisfaction of health status).
Author Response
Comments and Suggestions for Authors
The aim of this brief study was to evaluate the prevalence of frailty in Indonesian older adults as well as to describe the factors associated with the frailty. Here are my comments to the authors.
Abstract
I suggest to include the prevalence also of the data about pre-frailty people
Response: below is added
61.6% were prefrail
Line 27
Please delete “_”
Response: changed
Lines 45-50
I suggest to include the ethical approvals in the paper rather as citation.
Response: below is added
Ethics review boards of RAND and the University of Gadjah Mada in Indonesia approved the IFLS [19].
Lines 61-62
What type of Hand grip? Please add this information as well as the accuracy of the instrument
Response: below is added
Weakness was assessed with hand grip strength (HGS), using a “Baseline Smedley Spring type dynamometer” (calibrated daily), on each hand twice, using a HGS (kg) from all four measurements [24,29]. The Smedley dynamometer records measurements to the nearest 0.5 kg of force [30], the Pearson correlation between forces recorded with the Smedley dynamometer and known forces was .98 [31]
Line 65-66
IPAQ: the authors used the IPAQ to investigate PA levels. However the IPAQ is valid to assess PA in adults and not in older adults (aged over 60 years). Please discuss this aspect as limitation of the study. It is possible that the data may have a bias.
Response: below is added under study limitations
A further limitation was that the IPAQ [32,33] is used in populations 15-69 years, and in this study persons 70 years and older were included. In a validation study in Japan, the IPAQ was found a useful tool (adequate validity) for assessing physical activity among older adults [42]. However, some studies [e.g., 43] suggest to use a modified IPAQ for the elderly, which should be considered in future studies
Line 121
The data about region is missing
Response: added
it is divorced not divorved.
Response: Corrected
Economic background: based on table 1 the information is different. Please check “lower economic background”. The difference is between Poor and rich category.
Response: Corrected
Is lower economic background the same of poor in table 1? I suggest to use the same words. Please check all.
Response: Corrected
Line 142-143
What is the difference with the higher prevalence of the study conducted in Indonesia (reference 4)?
Please discuss and explain this aspect.
Response: added as below
A higher prevalence of frailty was found among geriatric clinic patients in Indonesia (25.2%) [3], which may be explained by the differences in the recruitment setting; a higher prevalence of frailty is expected in geriatric clinic patients at referral hospitals than in a community setting.
Line 171-172
I suggest to better explain the link between frailty, physical activity level and functional disability assessed by means of the ADL and IADL questionnaire. For example, it was demonstrated that frail individuals reported different levels of mobility function compared to no frail older adults (PMID: 30898096). Again a recent study suggested that frail elders, men, those who are older, overweight or have multiple comorbidities are most likely to have low activity (PMID: 29106478). I think that this is an important aspect of frailty and I suggested to the author to deepen this topics.
Response: added accordingly
Lines 175-178
Please explicit in the limitation of the study the use of single item to investigate some aspects about frailty (e.g., life satisfaction of health status).
Response: Added accordingly
Round 2
Reviewer 1 Report
The manuscript has been significantly improved, while whether the Poisson regression model suitable for this study should be explained and reviewed by a statistician.
Author Response
In analyses of data from cross-sectional studies, “Poisson models with robust variance are better alternatives than logistic regression is.” [40].
Reviewer 2 Report
I congratulate you for the important revision work. I think the article has improved in this form.
Author Response
there are no additional comments to respond to